# FACTUAL AND MUSICAL EVALUATION METRICS FOR MUSIC LANGUAGE MODELS

## ABSTRACT

Music language models (Music LMs), like vision language models, leverage multimodal representations to answer natural language queries about musical audio recordings. Although Music LMs are reportedly improving, we find that current evaluations fail to capture whether their answers are correct. Specifically, for all Music LMs that we examine, widely-used evaluation metrics such as BLEU, METEOR, and BERTScore fail to measure anything beyond linguistic fluency of the model's responses. To measure the true performance of Music LMs, we propose (1) a better general-purpose evaluation metric for Music LMs adapted to the music domain and (2) a factual evaluation framework to quantify the correctness of a Music LM's responses. Our framework is agnostic to the modality of the question-answering model and could be generalized to quantify performance in other open-ended question-answering domains. We use open datasets in our experiments and will release all code on publication.

## 1 INTRODUCTION

Music Language Models (Music LMs) are an emerging family of multimodal models that consume both language and audio as input. Music LMs answer natural language queries about music, making these models a new and promising general-purpose tool for music information retrieval tasks such as music captioning, music tagging, and interactive music question answering. Music LMs are typically benchmarked with Natural Language Processing (NLP) metrics such as BERTScore (Zhang et al., 2020), which compare reference text with model outputs using a question-answering (QA) dataset, e.g., MusicQA. Prior work has identified that these metrics may be inadequate (Gardner et al., 2024; Lee & Lee, 2024; Zang et al., 2025), but they remain the predominant approach for evaluating Music LMs.

In this work, we show that the standard NLP metrics used to assess Music LMs are not just inadequate; they fail to measure any ability of these models to extract information from audio. Specifically, we propose a baseline experiment that pairs each question in a Music QA dataset with a random, unrelated music recording from the dataset; this baseline tells us how a Music LM scores when it receives no useful information with which to answer the question; nevertheless, the standard NLP metrics judge outputs of this baseline to be equally good as when the correct music is provided. Furthermore, we show that adversarially crafted answers achieve very high scores under the standard metrics, despite being factually incorrect.

Given the shortcomings of standard NLP metrics, we propose two improvements to the Music LM evaluation protocols. First, we propose a new music-informed text evaluation metric, CLAPText, based on the pretrained CLAP embedding model (Wu et al., 2023). CLAPText is a simple drop-in replacement for pairwise NLP metrics within the standard evaluation framework. We find that CLAPText is a capable of judging a Music LM's use of audio information, in the sense that it prefers answers based on correct audio inputs over answers based on random audio inputs. Second, we propose a more interpretable *factual* evaluation framework for measuring specific aspects of musical understanding.

Our factual evaluation framework builds upon the work of Weck et al. (2024), which develops an new benchmark dataset for Music LMs based on multiple-choice question answering. In contrast to open-ended Music QA, multiple-choice question answering can be quantified via simple Precision, Recall, and F1 scores. We abstract and extend the procedure used by Weck et al. (2024) into a *framework*

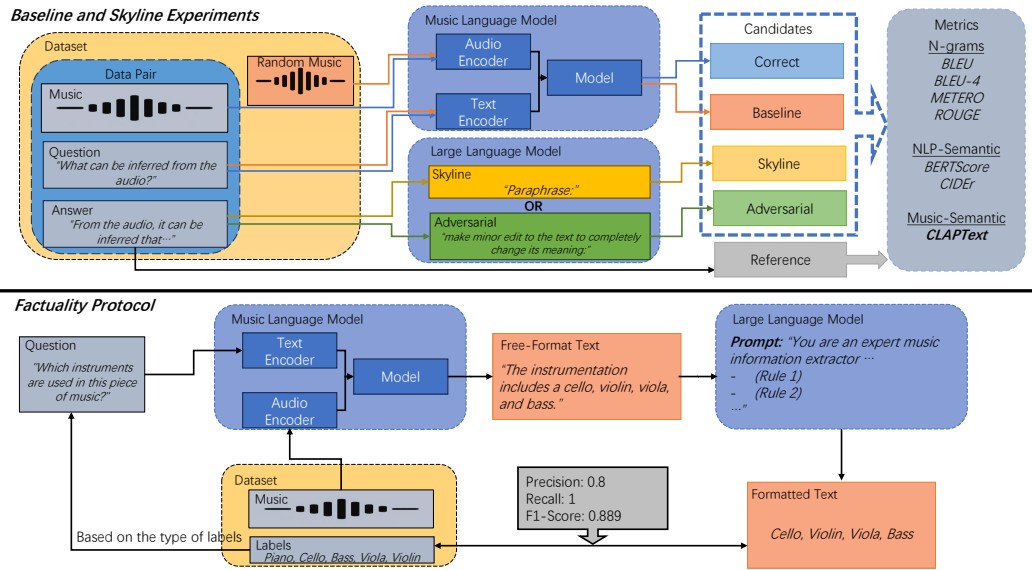

Figure 1: Overview of our evaluation methodology. **Top:** Given (question, audio) pairs, we study the behavior of open-ended text metrics by comparing reference answers to outputs of (1. Correct; blue) the Music LM, provided with the intended audio for the corresponding question; (2. Baseline; orange) the Music LM, provided an audio input chosen at random from the dataset; (3. Skyline; yellow) an LLM, asked to paraphrase the reference text; no Music LM should be able to outperform this skyline result (4. Adversarial; green) an LLM, asked to make subtle edits to the reference text that completely change its meaning. **Bottom:** Our factuality framework for converting a labeled dataset into a benchmark for Music LMs. A Music LM first predicts open-ended text in response to a prompt for factual information. A large language model then performs keyword extraction under strict rules to canonicalize this free-form response into structured labels. These extracted labels are compared to ground-truth labels to compute factuality metrics such as precision, recall, and F1-score, enabling direct, interpretable evaluation of factual correctness.

for transforming labeled datasets into factual evaluation benchmarks for Music LMs. The proposed framework is precise, granular, and cannot be 'fooled' by fluent but otherwise hallucinatory outputs from a Music LM. We implement examples of this framework using the Free Music Archive (FMA) (Defferrard et al., 2017) and MusicNet (Thickstun et al., 2017).

Our contributions are threefold:

- We show that six commonly reported metrics for evaluating Music LMs fail to measure these models' ability to extract information from audio inputs.
- We propose a new, musically-aware similarity metric, CLAPText: a drop-in replacement for the aforementioned metrics that more effectively quantifies Music LM performance.
- We propose and implement a modality-agnostic framework for measuring model *correctness*. Our framework uses a factual question-answering protocol, which can be evaluated using simple and interpretable Precision, Recall, and F1 scores.

Together, these findings highlight the limitations of current evaluation practice for Music LMs and address these limitations with new evaluation techniques along with evidence of their efficacy.

## 2 RELATED WORK

We study evaluation of Music LMs in the context of four distinct models, developed using a variety of architectures and training corpora. LTU-AS (Gong et al., 2023b) is a general-purpose audio language model finetuned from LLaMA-7B (Touvron et al., 2023). LLaMA-Adapter (Gao et al., 2023)

extends LLaMA-7B to multimodal inputs; in this work we use the ImageBind-LLM variant (Han et al., 2023), which embeds text, audio, video, and images in the ImageBind space (Girdhar et al., 2023). MU-LLaMA (Liu et al., 2023), also based on the LLaMA-Adapter architecture, is specialized for music captioning and QA, using MERT (Li et al., 2024) audio embeddings and trained on the MusicQA dataset. SALMONN (Tang et al., 2024), derived from Vicuna-7B (Zheng et al., 2023), combines Whisper and BEATs (Chen et al., 2022b) encodings through a Q-Former (Li et al., 2023) and is trained on a broad mix of music datasets; we evaluate the original audio-focused model rather than newer speech or video-specialized variants to keep comparisons consistent. Beyond the models evaluated in this work, LLark (Gardner et al., 2024) has demonstrated longer-form music captioning but lacks a public release, and the public checkpoint of MuMu-LLaMA (Liu et al., 2024) is corrupt.

Because Music LMs generate text-based responses, evaluation methods borrowed from the Natural Language Processing (NLP) literature are readily available and popular. Specifically, MU-LLaMA (Liu et al., 2023), LLark (Gardner et al., 2024), and MuMu-LLaMA (Liu et al., 2024) all compare generated answers to reference responses using BLEU (Papineni et al., 2002), BLEU-4, METEOR (Banerjee & Lavie, 2005), ROUGE, and BERTScore (Zhang et al., 2020). LLark adopts CIDEr (Vedantam et al., 2015) as an additional metric to evaluate music captioning. While these metrics are convenient, we will see in Section 3 that they are unable to assess the performance of Music LMs.

The most popular benchmark dataset for evaluating Music LMs—which we adopt in this work—is the MusicQA dataset, introduced in Liu et al. (2023). MusicQA consists of constructed question–answer pairs derived from a subset of MusicCaps (Agostinelli et al., 2023) (training data), MagnaTagATune (Wolff et al., 2012) (finetuning data), and MTG-Jamendo (Bogdanov et al., 2019) (test data). Beyond MusicQA, we apply new methods for evaluating the factuality of Music LM responses using the Free Music Archive (FMA) (Defferrard et al., 2017) and MusicNet (Thickstun et al., 2017).

## 3 FREE-FORM QUESTION ANSWERING

One may query a Music LM with musical audio, paired with a question about it, for example, *"what instrument is playing?"* The model responds in natural language but, importantly, this response has no imposed structure. Given the 'free-form' nature of the outputs, we call this setting Free-Form Question Answering (Free-Form QA). Past work measures performance on Free-Form QA using the NLP metrics previously described in Section 2. At first glance, NLP metrics appear to be a natural evaluation choice for free-form model outputs. In this section, we present a series of experiments to the contrary, and propose CLAPText as an alternative evaluation metric.

We conduct Free-Form QA experiments on the MusicQA dataset, which consists of (1) four generic captioning questions that apply to each audio example including "Describe the music," "Describe the music in detail," "What do you hear in the audio," and "What can be inferred from the audio," and (2) five audio-specific questions. An example music captioning question paired with its corresponding reference text included in the MusicQA dataset is given in Table 1.

### 3.1 BASELINE

To contextualize the Free-Form QA performance of Music LMs, we conduct a baseline experiment that pairs each question in the dataset with a random, unrelated music recording from the same dataset. This baseline tells us how models perform when they receive no useful information with which to answer a query; all Music LMs should outperform this baseline when they are provided the correct audio input. Towards a similar goal of decoupling auditory and linguistic reasoning, Zang et al. (2025) previously proposed replacing the audio component of a Music LM prompt with Gaussian noise, but they find this yields pathological performance degradation. We therefore prefer the in-distribution random sampling of alternative audio inputs over other audio corruption strategies.

### 3.2 SKYLINE (PARAPHRASE)

To establish an upper bound on NLP metric performance, we conduct a skyline experiment that emulates a near perfect response. To ensure that our idealized responses are correct but not identical to

Table 1: A sample from the MusicQA dataset that demonstrates our Free-Form QA transformations. In **bold** are the most musically relevant keywords. Observe that the Skyline's paraphrasing preserves keywords or substitutes them with synonyms (e.g. *raw → unpolished*, *experimental → exploratory*) while changing lexical structure. The Adversarial transformation does the opposite, dramatically changes the keywords but preserves the lexical structure.

| Source | Text |
|---|---|
| Reference Question | What can be inferred from the audio? |
| Reference Answer | From the audio, it can be inferred that the track is a blend of **post-rock** and **electronic experimental** sounds. The track features a variety of instrumentation, including **guitar, synthesizers, and samples**. The overall sound is **raw** and **experimental**, with a strong emphasis on **atmosphere and mood**. |
| Paraphrase | Based on the audio, the track appears to combine elements of **post-rock** and **electronic experimental** music. It includes diverse instrumentation such as **guitar, synthesizers, and samples**. The sound is **unpolished** and **exploratory**, focusing heavily on creating a particular **atmosphere and mood**. |
| Adversarial | From the audio, it can be inferred that the track is **purely classical with orchestral arrangements**. The track features **traditional instrumentation, including violin, cello, and piano**. The overall sound is **polished and structured**, with a strong emphasis on **melody and harmony**. |

the reference text, we paraphrase the reference text using `ChatGPT-4.1-mini`. The paraphrased response exhibits the correct answer, but written differently. Here we assume that current Large Language Models are capable of rephrasing language without changing its core meaning. We verified this by asking musicians to inspect and validate a sampling of paraphrased outputs, and we provide an example of the paraphrasing transformation in Table 1.

### 3.3 ADVERSARIAL

We consider an adversarial experiment to examine how the NLP metrics behave when for a response that is deliberately incorrect. Instead of asking `ChatGPT-4.1-mini` to preserve the meaning of the reference text, we ask it to *change* the meaning in as few edits as possible. This rewrite should therefore be lexically similar to the original, but carry different meaning. Changing the meaning renders the rewrite incorrect with respect to the original audio paired with it. A good evaluation metric should assign low scores to these adversarial answers. We provide an example of the adversarial transformation in Table 1.

### 3.4 CLAPTEXT

We propose CLAPText, a musically-aware semantic similarity metric that leverages pretrained CLAP (Wu et al., 2023) embeddings. Similar to how CLIP (Radford et al., 2021) contrastively learns a shared latent space between text and image, CLAP learns such a space for text and audio. We use the CLAP checkpoint trained on a combination of music, AudioSet, and LAION-Audio-630k with HTSAT (Chen et al., 2022a) to compute pairwise similarity scores that reflect the musical similarity between two pairs of text. Formally, CLAPText is defined as:

$$\text{CLAPText}(c, r) = s(\text{CLAP}(c), \text{CLAP}(r))$$

in which $c$ is the candidate text, $r$ is the reference text, $\text{CLAP}(x)$ is the CLAP embedding vector of some text input $x$, and $s(\cdot, \cdot)$ is the cosine similarity of the two embeddings. Conveniently, the inputs

Table 2: Aggregate similarity metrics between reference and response text on the MusicQA-Jamendo and MusicQA-MagnaTagATune Free-Form QA datasets. The datasets contain 5,040 and 70,011 QA pairs respectively. NLP metrics barely distinguish between the quality of answers to queries using the correct song as input, versus a random song from the dataset. We omit MagnaTagATune results for MU-LLaMA because the model is trained on this dataset. With the exception of CIDEr, all metrics are bound by 1.0. CIDEr in this context multiplies embedding similarity by 10, which bounds it within 0 and 10.

**MusicQA-Jamendo**

| Model | Prompt | BLEU | BLEU-4 | METEOR | ROUGE | BERTScore | CIDEr | CLAPText |
|---|---|---|---|---|---|---|---|---|
| LTU-AS | Correct | 0.2487 | 0.1643 | 0.2723 | 0.3144 | 0.8847 | 0.4731 | 0.4116 |
| | Random | 0.2505 | 0.1640 | 0.2749 | 0.3183 | 0.8863 | 0.4255 | 0.3534 |
| MU-LLaMA | Correct | 0.3015 | 0.2084 | 0.3891 | 0.4609 | 0.8997 | 0.3288 | 0.5282 |
| | Random | 0.2906 | 0.1961 | 0.3779 | 0.4529 | 0.8968 | 0.2858 | 0.4514 |
| LLaMA Adapter | Correct | 0.2001 | 0.1321 | 0.3270 | 0.5201 | 0.8915 | 0.1063 | 0.4426 |
| | Random | 0.1951 | 0.1260 | 0.3163 | 0.5096 | 0.8889 | 0.1013 | 0.3797 |
| SALMONN | Correct | 0.2950 | 0.2197 | 0.3505 | 0.4184 | 0.8985 | 0.9262 | 0.5066 |
| | Random | 0.2700 | 0.2041 | 0.3270 | 0.3836 | 0.8918 | 0.9232 | 0.4050 |
| | Paraphrase | 0.5956 | 0.4648 | 0.5968 | 0.5663 | 0.9581 | 1.7632 | 0.8233 |
| | Adversarial | 0.7413 | 0.6614 | 0.7739 | 0.7609 | 0.9608 | 3.8270 | 0.5712 |

**MusicQA-MagnaTagATune**

| Model | Prompt | BLEU | BLEU-4 | METEOR | ROUGE | BERTScore | CIDEr | CLAPText |
|---|---|---|---|---|---|---|---|---|
| LTU-AS | Correct | 0.2698 | 0.1884 | 0.3320 | 0.3914 | 0.9015 | 0.6085 | 0.4475 |
| | Random | 0.2524 | 0.1712 | 0.3138 | 0.3717 | 0.8971 | 0.5253 | 0.3431 |
| LLaMA Adapter | Correct | 0.3009 | 0.2208 | 0.3795 | 0.4707 | 0.9098 | 0.8840 | 0.4754 |
| | Random | 0.2831 | 0.2048 | 0.3657 | 0.4646 | 0.9057 | 0.7958 | 0.3728 |
| SALMONN | Correct | 0.3109 | 0.2526 | 0.3869 | 0.4563 | 0.9074 | 1.4046 | 0.5326 |
| | Random | 0.2950 | 0.2354 | 0.3679 | 0.4376 | 0.9026 | 1.2886 | 0.4103 |
| | Paraphrase | 0.5622 | 0.4359 | 0.6137 | 0.5738 | 0.9596 | 1.5311 | 0.8137 |
| | Adversarial | 0.7597 | 0.6891 | 0.8019 | 0.7937 | 0.9682 | 3.8919 | 0.5884 |

to CLAPText are text pairs just like the NLP metrics, making it a drop-in replacement or addition to existing evaluation pipelines with minimal code changes.

Intuitively, CLAPText measures the semantic similarity between candidate and reference answers in an embedding space explicitly trained on music-text pairs. We hypothesize that this allows CLAP-Text to capture music-specific semantics better lexical NLP metrics and more general semantic similarity scores like BERTScore.

## 3.5 RESULTS

We report NLP metric values and our proposed CLAPText metric for the Free-Form QA performance of several contemporary Music LMs in Table 2. We summarize our findings below.

**Correct is hardly better than Random**. Recall that 'Correct' here means querying the Music LM as intended with the unaltered validation prompts. This is representative performance of the model with no tricks whatsoever. For all metrics except CLAPText, the maximum difference between the correct and random baseline score in any experiment is CIDEr with 0.1106. This is very small considering CIDEr's maximum value is 10. Excluding CIDEr, the score with the second largest discrepancy in similarity between intended use and random choice is ROGUE, with a max margin of 0.03472. Interestingly, the random choice prompt occasionally achieves better performance than the unaltered one. This occurs frequently for the LTU-AS model in the MusicQA-Jamendo dataset - take note of BLEU, METEOR, ROGUE, and BERTScore in particular.

**Adversarial Crosses Skyline consistently**. With the exception of CLAPText, every metric assigns a higher score to the Adversarial prompt than the Skyline; this is the reverse of the expected and reasonable order. Our design of the Adversarial prompt is effective in 'fooling' these metrics, as it is lexically similar to the reference text by construction, yet is factually incorrect. The Skyline scores are also weak; despite being a paraphrasing of the reference text, the metrics that are bound within $[0, 1]$ hover within approximately the 0.4-0.6 range.

**BERTScore assigns very high similarity**. The lowest aggregate similarity between any response and the reference text assigned by BERTScore is 0.8847 for the unaltered ('Original') prompt strategy of LTU-AS on the MusicQA-Jamendo dataset. As mentioned earlier, this score is lower in similarity with the reference text than are the responses elicited by the random choice baseline. Within the bounds of $[0, 1]$, 0.8847 is quite high considering it is the lowest BERTScore we obtained. It could be that the most important and distinguishing musical semantics in the responses are diluted by boilerplate text. While the deep embeddings underpinning BERTScore are powerful in general language settings, the non-discriminative similarities we observe here demonstrate that we need embeddings that reflect both the text-audio multimodality and musical semantics.

**CLAPText is the only metric that correctly orders the Adversarial and Skyline**. Encouragingly, CLAPText is the only metric that is not 'fooled' by linguistic similarity and can discern musical inconsistency. CLAPText always assigns higher similarity scores to the paraphrased Skyline than it does to the falsified Adversarial. Recall that the Adversarial prompt is deliberately made incorrect, so it *should* be misaligned with the reference text and be assigned a low score.

Our findings reveal fundamental limitations of many commonly reported NLP metrics for music-language understanding. Evaluations that rely solely on these metrics risk misrepresenting model capabilities and progress. The CLAPText metric shows promise as a drop-in alternative towards higher quality performance measurement. That said, we can get even more granular, factual assessment by extracting discrete labels from free-form responses. To more directly assess whether models are *correct* about the music we give them, we propose a complementary factual evaluation *framework* in Section 4. Our framework enables clearer comparisons between ground-truth music annotations and key aspects of model understanding.

## 4 FACTUAL QUESTION ANSWERING

In light of difficulties evaluating Free-Form QA, we propose a targeted evaluation framework for probing Music LMs on matters of factuality (Factual QA), for example, genre classification or instrument recognition. In principle, these questions can be evaluated using simple metrics, e.g., accuracy. In practice, music language models produce free-form text, requiring analysis to determine if their response is correct. To solve this problem, we propose to use a strong language model (in our case, `ChatGPT-4.1-mini`) to parse the music language model's output and extract a structured response, e.g., a list of labels in some closed vocabulary. The general structure of the factuality protocol is shown in the lower part of Figure 1.

Our evaluation framework proceeds in three steps. First, for each audio recording in the dataset, we ask the Music LM a factual question tailored to the dataset's labels. Second, we apply a structured keyword extraction protocol (detailed in Section 4.1) to convert the model's unconstrained textual output into a canonical list of labels drawn from the dataset-specific vocabulary of labels. Finally, we compare these extracted labels against the ground-truth labels provided by dataset. By aligning both model predictions and dataset labels into the same structured form, we can evaluate factual correctness with standard metrics such as Accuracy, Precision, Recall, and F1.

In the remainder of this section, we present this protocol in full detail. Our approach consists of three components: converting free-form outputs into structured representations through a keyword extraction protocol (Section 4.1); designing evaluation pipelines that account for both chunked and unchunked model architectures (Section 4.2); and analyzing results across multiple task formulations and prompting strategies (Section 4.3). Together, these elements establish a principled and interpretable evaluation methodology for Factual QA with Music LMs.

Table 3: An example for multiple keywords extraction. In the table, MU-LLaMA gives two possible answers to the question about genre without preference. Here we accept all keywords generated by the model and compare them to the ground truth in precision/recall/F1-score. Note that the chunk size of MU-LLaMA/LLaMA-Adapter is longer than the audio file in FMA, so there is no difference between chunked models and unchunked models for the genre classification task on FMA. In this example, Precision = 1, Recall = 0.5, F1-Score = 0.667

| Source | Text |
|---|---|
| Factual Question | What genre does this piece of music fall under? |
| Ground Truth | Pop |
| Model (MU-LLaMA) | This piece of music falls under the genre of pop/soft rock. |
| Extracted Labels | Pop, Rock |

## 4.1 KEYWORD EXTRACTION

The cornerstone of our evaluation protocol is the conversion of free-form text into a canonical structured form that can be compared directly with ground-truth labels. To minimize the confounding influence of natural language surface form, we employ a keyword extraction step using `ChatGPT-4.1-mini`, a high-performing general-purpose language model. Importantly, this step does not attempt to infer or interpret beyond what is explicitly stated; rather, it enforces a strict set of rules to ensure consistent, reproducible, and conservative extraction.

Our objective is not to infer labels from stylistic cues, but to convert *explicitly stated* labels into a structured, machine-checkable form so that simple metrics (Accuracy, Precision/Recall/F1) can be computed against a closed vocabulary. For example, if the model writes, "*The genre of the song is rock*," the extractor returns *rock*; if it writes, "*Instruments: double bass and horns*," the extractor returns *bass, horn* after canonicalization. By restricting extraction to exact mentions (with light normalization), we avoid over-crediting implied associations while also preventing under-crediting due to surface-form variation, enabling consistent, automatable evaluation. The specific rules we obey to contract prompt for LLM is provided in Appendix B.1.

Dataset annotations are correspondingly normalized to match these canonical forms. For the instrumentation task, we observed significant inconsistencies in human annotation and model phrasing (e.g., "double bass" vs. "contrabass"). To mitigate spurious mismatches, we apply a post-extraction normalization filter: all piano variants are mapped to *piano*, all horn variants to *horn*, and both contrabass and double bass to *bass*. Additionally, all labels are lowercased to eliminate differences due to capitalization. No such normalization is applied for genre classification, where small differences in descriptors often reflect meaningful distinctions, nor for composer classification, where the extraction model is explicitly instructed to return the simplest widely recognized form of each composer's name.

## 4.2 EVALUATION METRICS FOR KEYWORD LABELS

A central complication in factuality evaluation is that models may output multiple answers even for questions that have only a single ground-truth label. This behavior makes Accuracy, which assumes one-to-one correspondence, an unreliable evaluation metric. Instead, we adopt Precision, Recall, and F1 Score, which allow us to treat model predictions as sets of labels and measure both correctness and over-generation. One contributing factor is architectural: chunked models, which process audio in 60-second segments, naturally produce multiple outputs across different chunks. Even unchunked full-length models sometimes hedge their responses, giving several possible answers. Meanwhile, unchunked models may also give ambiguous output within which multiple guesses for the answer are given without specific preference. An example is given in Table 3.

In summary, the tendency of models to produce multiple predictions—even in tasks with a single ground-truth label—necessitates evaluating with Precision, Recall, and F1 rather than Accuracy alone. This design ensures that evaluation reflects both correctness and over-generation, while avoiding artificial penalties on models that provide more than one plausible answer.

Table 4: Factual QA for instrument recognition and genre classification, using two different prompts for each task. With a good prompt, models perform much better than chance (the Random baseline) but are far from perfect (F1-Score = 1). Explicitly enumerating the list of possible instruments or genres (indicated by {*}) in the prompt confuses every tested music language model.

**Prompt: "Which instruments are used in this piece of music?"**

| | LTU-AS | | MU-LLaMA | | LLaMA-Adapter | |
| | Correct | Random | Correct | Random | Correct | Random |
|---|---|---|---|---|---|---|
| Precision | 0.534 | 0.364 | 0.367 | 0.266 | 0.488 | 0.350 |
| Recall | 0.461 | 0.314 | 0.582 | 0.422 | 0.676 | 0.486 |
| F1-Score | 0.495 | 0.337 | 0.450 | 0.326 | 0.567 | 0.407 |

**Prompt: "Among {*}, which instruments are used in this piece of music?"**

| | LTU-AS | | MU-LLaMA | | LLaMA-Adapter | |
| | Correct | Random | Correct | Random | Correct | Random |
|---|---|---|---|---|---|---|
| Precision | 0.155 | 0.149 | 0.199 | 0.172 | 0.164 | 0.164 |
| Recall | 0.714 | 0.689 | 0.773 | 0.668 | 0.923 | 0.920 |
| F1-Score | 0.254 | 0.245 | 0.316 | 0.273 | 0.279 | 0.278 |

**Prompt: "What genre does this piece of music fall under?"**

| | MU-LLaMA | | LLaMA-Adapter | | SALMONN | |
| | Correct | Random | Correct | Random | Correct | Random |
|---|---|---|---|---|---|---|
| Precision | 0.256 | 0.102 | 0.334 | 0.081 | 0.293 | 0.084 |
| Recall | 0.291 | 0.115 | 0.342 | 0.083 | 0.388 | 0.111 |
| F1-Score | 0.272 | 0.108 | 0.338 | 0.082 | 0.334 | 0.096 |

**Prompt: "Among {*}, what is the genre of this song?"**

| | MU-LLaMA | | LLaMA-Adapter | | SALMONN | |
| | Correct | Random | Correct | Random | Correct | Random |
|---|---|---|---|---|---|---|
| Precision | 0.201 | 0.124 | 0.333 | 0.111 | 0.179 | 0.124 |
| Recall | 0.188 | 0.116 | 0.327 | 0.109 | 0.264 | 0.183 |
| F1-Score | 0.195 | 0.120 | 0.330 | 0.110 | 0.213 | 0.148 |

## 4.3 RESULTS

We implement Factual QA for two representative music understanding tasks: *instrumentation recognition* and *genre classification*; results are presented in Table 4. For genre classification, we use the FMA-Small subset of FMA, consisting of 8,000 thirty-second clips evenly distributed across eight top-level genres: hip-hop, pop, folk, experimental, rock, international, electronic, and instrumental (1,000 clips per genre). FMA also provides a hierarchical taxonomy for genre classification, in which each top-level genre contains multiple subgenres organized in a tree structure; we only use top-level genre labels (see Appendix C.3 for an application of our framework to full genre trees in FMA). For instrument recognition, we use the MusicNet dataset, consisting 330 full-length classical recordings with detailed annotations, including a composer label for each recording (one of ten composers) and a list of instruments represented in each recording.

To measure the impact of prompt engineering, we conduct our experiment with multiple prompting strategies. Some prompting strategies are taken directly from the models' demo pages to reflect the linguistic patterns in their training data, while others are constructed by us to cover a broader range of language styles. In the main paper we report performance under two prompting settings: the best-performing prompt for each model, and a prompt that explicitly lists all possible answers (which we initially hypothesized would be easier). It is worth noting that the prompts yielding the highest performance are not always those recommended in the official demo pages. An elaboration upon different prompting strategies is provided in Appendix B.2.

Unlike standard NLP metrics, our factuality metrics clearly distinguish between the correct audio experiment and the random audio baseline. The differentiation is especially pronounced in the genre classification task, where the F1 score shows a significant gap between correct-song and random-song baselines. This is likely because the wide coverage of musical genres in FMA appears to overlap more with model training data, which may explain stronger performance. In contrast, instrumentation classification shows a smaller gap between correct and random baselines, though correct-song results are consistently better. Models perform reliably on common instruments such as piano and violin but struggle with less frequent ones such as oboe or harpsichord, likely reflecting the limited representation of these instruments in pretraining data. Results on the MusicNet composer classification task, reported in Appendix C.2, are overall worse, which is consistent with our expectation that classical composers are underrepresented in model training corpora.

For instrument recognition, we experimented with a prompting strategy that provides the list of possible instruments explicitly in the prompt. We expected this strategy to simplify the Music LM's task by informing the model that the output label should be one of provided options, thus turning an open-ending question into a choice between the provided labels. However, we observe empirically that this prompting strategy often confuses the models. For instrumentation, models tend to output many of the provided labels, resulting in high recall scores but very low precision: most correct instruments are included, but predictions also contains many false positives. This phenomenon is less pronounced for classification tasks but still degrades performance, as seen in both genre and composer experiments.

For genre classification, it is noteworthy that there are eight possible genre labels, distributed evenly over the FMA-Small dataset; a model that randomly guesses genre labels should score around $0.125$. Nevertheless, the baseline random audio experiment scores slightly lower than $0.125$ because the models sometimes respond with no answer, or equivocate among multiple answers. Like instrument recognition, when we attempt to provide the list of genre options explicitly to the Music LMs, performance suffers. We further tested alternative prompting formats such as true–false and multiple-choice prompts, which also prove to confuse the models (see Appendix B.2).

In summary, our results demonstrate that while music-language models capture some factual properties of music, their factuality performance lags behind what standard NLP metrics suggest. Furthermore, model outputs are highly sensitive to prompting, underscoring the fragility of current approaches.

## 5 CONCLUSION

We find that existing evaluation metrics for Music LMs place disproportionate emphasis on the surface form of language—rewarding stylistic fluency, lexical overlap, and generic semantic similarity—while placing surprisingly little weight on the factual comprehension of music. As a result, current evaluation practices may inadvertently steer the development of music–language models toward producing text that *sounds* right rather than text that *is* right. Evaluations must be able to discriminate between outputs that are merely well-phrased and those that convey musically accurate insights. Our proposed CLAPText metric is a musically-informed, drop-in replacement for existing evaluation metrics, and our proposed Factual QA protocol offers a more fine-grained analysis of the capabilities and deficiencies of Music LMs.

Encouragingly, our Factual QA experiments suggest that current Music LMs already exhibit some capacity for factual comprehension, even if this ability is under-rewarded by standard NLP metrics. By asking unambiguous, content-grounded questions, and evaluating responses using metrics sensitive to factual correctness rather than linguistic similarity, we demonstrate one way to realign evaluation towards the goal of reliability and accuracy. We hope this framework can provide a clearer feedback loops for model developers to improve the capabilities of their models. Ultimately, building reliable multimodal language models—whether for music, science, or other domains—will require evaluation metrics that reward factual understanding as much as stylistic fluency. We hope that the Factual QA framework described in this paper for evaluating Music LMs can be extended beyond the music domain and facilitate the development of reliable linguistic information retrieval models in other modalities.

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
