., 2023) for a variety of QA tasks including speech, music, and sound effects. LTU-AS processes audio inputs tokenized using Whisper (Radford et al., 2022) and TLTR (Gong et al., 2023a) encoders, and is trained on the Open-ASQA dataset developed by the LTU-AS authors. This dataset aggregates samples from multiple sources, including AS-Strong, AudioSet, VGGSound, FSD50K, AudioCaps, FreeSound, Clotho, SoundBible, IEMOCAP, LibriTTS, VoxCeleb2, MOSEI, and FMA.

LLaMA-Adapter(Gao et al., 2023) is a general-purpose multimodal language model finetuned from LLaMA-7B and designed to target a variety of modalities including audio and, specifically, music. In this paper, we use ImageBind-LLM(Han et al., 2023), one of the latest variants of LLaMA-Adapter. ImageBind-LLM tokenizes text, audio, video, and images into a shared ImageBind embedding space (Girdhar et al., 2023). ImageBind itself is trained on a combination of datasets, including AudioSet, ESC (5-fold), Clotho, AudioCaps, and VGGSound.

MU-LLaMA(Liu et al., 2023) is designed specifically for music captioning and music question–answering tasks. It follows the LLaMA-Adapter(Gao et al., 2023) architecture derived from LLaMA-7B, but replaces Whisper-based audio tokenization with MERT embeddings (Li et al., 2024). MU-LLaMA is trained, fine-tuned, and evaluated using the MusicQA dataset developed by its authors.

SALMONN(Tang et al., 2024) is a general-purpose audio language model that processes audio tokenized using Whisper and BEATs(Chen et al., 2022b) representations fused according to the Q-Former architecture (Li et al., 2023). While multiple variants of SALMONN exist, including newer versions specialized for video and speech, in our experiments we select the original version of SALMONN to maintain focus on audio modeling. SALMONN is derived from the Vicuna-7B fine-tuned variant of LLaMA-2 7B (Zheng et al., 2023). It is trained on a diverse set of audio datasets, including LibriSpeech, GigaSpeech, CoVoST2-En2Zh, AudioCaps, Clotho, IEMOCAP, MusicCaps, LibriMix, VoxCeleb1, WavCaps, MillionSong, and MusicNet.

Beyond these widely tested models, other systems have been proposed in the literature. LLark (Gardner et al., 2024) enables more detailed music captioning on longer paragraphs, but has not released checkpoints for replication. MuMu-LLaMA (Liu et al., 2024) is another recent model, though the currently available checkpoint is corrupted and cannot be systematically tested. As more trained models and usable checkpoints are released, the evaluation frameworks established in prior work will provide a foundation for more comprehensive assessment of factuality and generative quality in music–language modeling.

## B    FULL FACTUALITY EXPERIMENT

### B.1    KEYWORD EXTRACTION RULES

We define the following rules for factual keyword extraction, which are consistently applied across all tasks and categories:

1. **Exact mention requirement:** Only terms that explicitly appear in the model's output are returned.

2. **No guessing:** Implicit inference or contextual associations (e.g., "jazz" inferred from "swing rhythm") are excluded.

3. **Comparatives:** For comparative statements ("X more than Y"), only the preferred entity (X) is retained.

4. **Deduplication and ordering:** Duplicate mentions are removed, while the order of first appearance is preserved.

5. **Output format:** Results are returned as a comma-separated list, e.g., *rock, jazz, classical*.

Table 5: Different Prompts of Instrument Recognition Experiments on MusicNet

**Prompt: "Which instruments are used in this piece of music?"**

| Metric | LTU-AS | | MU-LLaMA | | LLaMA-Adapter | |
|---|---|---|---|---|---|---|
| | Correct | Random | Correct | Random | Correct | Random |
| Precision | 0.534 | 0.364 | 0.367 | 0.266 | 0.488 | 0.350 |
| Recall | 0.461 | 0.314 | 0.582 | 0.422 | 0.676 | 0.486 |
| F1-Score | 0.495 | 0.337 | 0.450 | 0.326 | 0.567 | 0.407 |

**Prompt: "Which instruments constitute the instrumentation of this piece?"**

| Metric | LTU-AS | | MU-LLaMA | | LLaMA-Adapter | |
|---|---|---|---|---|---|---|
| | Correct | Random | Correct | Random | Correct | Random |
| Precision | 0.502 | 0.316 | 0.302 | 0.246 | 0.458 | 0.325 |
| Recall | 0.518 | 0.327 | 0.584 | 0.474 | 0.733 | 0.520 |
| F1-Score | 0.510 | 0.321 | 0.398 | 0.324 | 0.564 | 0.400 |

**Prompt: "What instruments are playing in this song?"**

| Metric | LTU-AS | | MU-LLaMA | | LLaMA-Adapter | |
|---|---|---|---|---|---|---|
| | Correct | Random | Correct | Random | Correct | Random |
| Precision | 0.532 | 0.334 | 0.271 | 0.211 | 0.472 | 0.357 |
| Recall | 0.423 | 0.266 | 0.544 | 0.423 | 0.643 | 0.486 |
| F1-Score | 0.472 | 0.296 | 0.362 | 0.282 | 0.545 | 0.411 |

**Prompt: "Among {all instruments}, which instruments are used in this piece of music?"**

| Metric | LTU-AS | | MU-LLaMA | | LLaMA-Adapter | |
|---|---|---|---|---|---|---|
| | Correct | Random | Correct | Random | Correct | Random |
| Precision | 0.155 | 0.149 | 0.199 | 0.172 | 0.164 | 0.164 |
| Recall | 0.714 | 0.689 | 0.773 | 0.668 | 0.923 | 0.920 |
| F1-Score | 0.254 | 0.245 | 0.316 | 0.273 | 0.279 | 0.278 |

6. **Empty case:** If no relevant term is mentioned, the output is an empty string.

7. **Canonical form:** All terms are normalized to simplified canonical forms (e.g., "J.S. Bach" → "Bach", "Acoustic Grand Piano" → "piano").

8. **No stylistic inclusion:** Descriptions of mood or affect without explicit mention (e.g., "a jazz-like feeling") are ignored.

## B.2 RESULTS

In our factuality experiments, we explored multiple prompting strategies to minimize the influence of wording on model performance. For regular question types, we varied the linguistic style of the prompts across casual, professional, and colloquial formulations. The results of these comparisons are reported in Table 5, and 6. We also tested a setting in which all possible answers from the dataset were explicitly provided within the prompt; however, this approach did not yield improvements over the regular prompting strategy.

To further simplify evaluation in a human-interpretable way, we designed two additional formats. For classification tasks, we adopted a binary-choice format, where the model was asked to choose between the exact ground-truth answer and one randomly selected distractor, framed as: "Between A and B, …". To mitigate positional bias, the order was randomized such that the correct answer appeared first in 50% of the prompts and second in the remaining 50%. The result of binary-choice experiments on MusicNet and FMA is shown in Table 13 and Table 7.

For recognition tasks, we asked the model to make a true–false judgment for each possible label in the dataset, producing a table of boolean values indicating whether the model believed each label was present in the given audio. From these predictions, we computed accuracy, which accounts for both true positives and true negatives, thereby reflecting the model's correctness on a label-by-label

Table 6: Different Prompts of Genre Classification (Same Tree) Experiments on FMA

**Prompt: "What is the genre of this song"**

| Metric | MU-LLaMA | | LLaMA-Adapter | | SALMONN | |
|---|---|---|---|---|---|---|
| | Correct | Random | Correct | Random | Correct | Random |
| Precision | 0.269 | 0.098 | 0.331 | 0.095 | 0.467 | 0.109 |
| Recall | 0.306 | 0.112 | 0.339 | 0.097 | 0.220 | 0.051 |
| F1-Score | 0.286 | 0.105 | 0.335 | 0.096 | 0.299 | 0.069 |

**Prompt: "What can you infer about the genre of the music"**

| Metric | MU-LLaMA | | LLaMA-Adapter | | SALMONN | |
|---|---|---|---|---|---|---|
| | Correct | Random | Correct | Random | Correct | Random |
| Precision | 0.218 | 0.086 | 0.272 | 0.077 | 0.256 | 0.083 |
| Recall | 0.364 | 0.143 | 0.407 | 0.115 | 0.389 | 0.126 |
| F1-Score | 0.273 | 0.107 | 0.326 | 0.092 | 0.309 | 0.100 |

**Prompt:"What genre does this piece of music fall under?"**

| Metric | MU-LLaMA | | LLaMA-Adapter | | SALMONN | |
|---|---|---|---|---|---|---|
| | Correct | Random | Correct | Random | Correct | Random |
| Precision | 0.256 | 0.102 | 0.334 | 0.081 | 0.293 | 0.084 |
| Recall | 0.291 | 0.115 | 0.342 | 0.083 | 0.388 | 0.111 |
| F1-Score | 0.272 | 0.108 | 0.338 | 0.082 | 0.334 | 0.096 |

**Prompt: "Among {all genres}, what is the genre of this song?"**

| Metric | MU-LLaMA | | LLaMA-Adapter | | SALMONN | |
|---|---|---|---|---|---|---|
| | Correct | Random | Correct | Random | Correct | Random |
| Precision | 0.201 | 0.124 | 0.333 | 0.111 | 0.179 | 0.124 |
| Recall | 0.188 | 0.116 | 0.327 | 0.109 | 0.264 | 0.183 |
| F1-Score | 0.195 | 0.120 | 0.330 | 0.110 | 0.213 | 0.148 |

Table 7: Binary Choices Experiment of Genres on FMA

**Prompt: "Between {two genres}, what is a better description of the genre of this piece?"**

| Metric | MU-LLaMA | LLaMA-Adapter | SALMONN |
|---|---|---|---|
| Precision | 0.464 | 0.500 | 0.460 |
| Recall | 0.740 | 0.590 | 0.853 |
| F1-Score | 0.570 | 0.542 | 0.598 |

basis. For chunked models, since they may give different opinions on different chunks of the music, we employed two strategies:

1. If the model returns true on any of the chunks, we take the final response as true. This strategy is to ensure that, for instruments that only appears in a short section of the whole music, the final output is still be able to reflect if the model detects them. The corresponding result is shown in Table 8.

2. If the model returns true on the majority of the chunks, we take the final response to be true and vise versa. This strategy is to show if the model has tenancy towards one of the answers instead of random guesses. The corresponding result is shown in Table 9.

## C MORE EXPERIMENTS

### C.1 SKYLINE AND BASELINE EXPERIMENTS ON MUSIC CAPTIONING TASKS

As we mentioned in 2, there are 4 music captioning questions corresponding to every audio file in MusicQA dataset with their associated answers. In the paper, we have reported the result for the ex-

Table 8: True-False Experiments of Instrumentations on MusicNet (Included)

**Prompt: Is {instrument} used in this song?**

| Metric | LTU-AS | | MU-LLaMA | | LLaMA-Adapter | |
|---|---|---|---|---|---|---|
| | Correct | Wrong | Correct | Wrong | Correct | Wrong |
| Precision | 0.182 | 0.171 | 0.179 | 0.171 | 0.179 | 0.174 |
| Recall | 0.670 | 0.628 | 0.876 | 0.839 | 0.876 | 0.854 |
| F1-Score | 0.287 | 0.268 | 0.297 | 0.284 | 0.297 | 0.289 |
| Accuracy | 0.167 | 0.149 | 0.168 | 0.167 | 0.174 | 0.166 |

Table 9: True-False Experiments of Instrumentations on MusicNet (Majority)

**Prompt: Is {instrument} used in this song?**

| Metric | LTU-AS | | MU-LLaMA | | LLaMA-Adapter | |
|---|---|---|---|---|---|---|
| | Correct | Wrong | Correct | Wrong | Correct | Wrong |
| Precision | 0.181 | 0.166 | 0.190 | 0.191 | 0.192 | 0.189 |
| Recall | 0.659 | 0.607 | 0.601 | 0.605 | 0.795 | 0.780 |
| F1-Score | 0.284 | 0.261 | 0.288 | 0.290 | 0.310 | 0.304 |
| Accuracy | 0.165 | 0.152 | 0.168 | 0.164 | 0.183 | 0.176 |

Table 10: Original Song and Random Song Experiment on Music Captioning Subset of Questions in MusicQA-Jamendo (comparable to the LLark experimental protocol).

| Model | Input | BLEU | BLEU-4 | METEOR | ROUGE | BERTScore | CIDEr |
|---|---|---|---|---|---|---|---|
| LTU-AS | Correct | 0.1599 | 0.0728 | 0.1456 | 0.1835 | 0.8565 | 0.0161 |
| | Random | 0.1538 | 0.0684 | 0.1435 | 0.1820 | 0.8557 | 0.0117 |
| MU-LLaMA | Correct | 0.2710 | 0.1575 | 0.2833 | 0.3448 | 0.8881 | 0.0736 |
| | Random | 0.2539 | 0.1396 | 0.2649 | 0.3271 | 0.8836 | 0.0515 |
| LLaMA Adapter | Correct | 0.1889 | 0.1008 | 0.2389 | 0.3894 | 0.8733 | 0.0165 |
| | Random | 0.1780 | 0.0889 | 0.2190 | 0.3670 | 0.8682 | 0.0112 |
| SALMONN | Correct | 0.1738 | 0.0873 | 0.2046 | 0.3199 | 0.8729 | 0.0145 |
| | Random | 0.1661 | 0.0778 | 0.1915 | 0.3028 | 0.8682 | 0.0116 |
| Rewrite | Skyline | 0.6040 | 0.4624 | 0.5810 | 0.5567 | 0.9571 | 1.5214 |
| | Adversarial | 0.7028 | 0.6023 | 0.7317 | 0.7143 | 0.9544 | 2.8867 |

periments on the whole MusicQA dataset. We also conducted the baseline and skyline experiments on the 4 music captioning questions. The result is shown in Table 10 and Table 11, which still supports our main conclusion in the paper. The only difference from the result on the entire MusicQA dataset is the drastic drop of CIDEr Score. In fact, the range of CIDEr score here is consistent with results reported by (Gardner et al., 2024), which is also evaluated on music captioning tasks. Our assessment is that CIDEr Score tends to return a low score on free-form music captioning tasks.

## C.2 COMPOSER EXPERIMENT ON MUSICNET

We also leverage the composer annotations in the MusicNet dataset to evaluate models on classical music. Since the training sets of the tested models do not contain classical repertoire, this task provides a challenging out-of-domain evaluation. As shown in Table 12 and Table 13, model performance on composer identification is notably weaker compared to the other two main experiments reported in the paper, indicating that current music–language models struggle with domains absent from their training data.

Table 11: Correct Song and Random Song Experiment on Music Captioning Subset of Questions in MusicQA-MagnaTagATune (comparable to the LLark experimental protocol).

| Model | Input | BLEU | BLEU-4 | METEOR | ROUGE | BERTScore | CIDEr |
|---|---|---|---|---|---|---|---|
| LTU-AS | Correct | 0.1674 | 0.0801 | 0.1768 | 0.2259 | 0.8736 | 0.0303 |
| | Random | 0.1528 | 0.0676 | 0.1621 | 0.2094 | 0.8685 | 0.0142 |
| LLaMA | Correct | 0.1943 | 0.0992 | 0.2310 | 0.3414 | 0.8857 | 0.0615 |
| Adapter | Random | 0.1796 | 0.0864 | 0.2164 | 0.3305 | 0.8808 | 0.0403 |
| SALMONN | Correct | 0.1177 | 0.0614 | 0.1884 | 0.3277 | 0.8774 | 0.0245 |
| | Random | 0.1061 | 0.0492 | 0.1685 | 0.3070 | 0.8707 | 0.0103 |
| Rewrite | Skyline | 0.5654 | 0.4297 | 0.5940 | 0.5701 | 0.9608 | 1.2813 |
| | Adversarial | 0.7339 | 0.6464 | 0.7722 | 0.7607 | 0.9646 | 2.9677 |

Table 12: Different Prompts of Composer Classification Experiments on MusicNet

**Prompt: "Which classical composer's style does this piece resemble the most?"**

| Metric | LTU-AS | | MU-LLaMA | | LLaMA-Adapter | |
|---|---|---|---|---|---|---|
| | Correct | Random | Correct | Random | Correct | Random |
| Precision | 0.232 | 0.238 | 0.181 | 0.179 | 0.268 | 0.240 |
| Recall | 0.239 | 0.245 | 0.424 | 0.418 | 0.497 | 0.445 |
| F1-Score | 0.236 | 0.242 | 0.254 | 0.250 | 0.348 | 0.312 |

**Prompt: "Which classical composer's compositional style does this piece most closely resemble?"**

| Metric | LTU-AS | | MU-LLaMA | | LLaMA-Adapter | |
|---|---|---|---|---|---|---|
| | Correct | Random | Correct | Random | Correct | Random |
| Precision | 0.297 | 0.334 | 0.156 | 0.157 | 0.316 | 0.279 |
| Recall | 0.261 | 0.294 | 0.348 | 0.352 | 0.521 | 0.461 |
| F1-Score | 0.277 | 0.313 | 0.215 | 0.217 | 0.394 | 0.348 |

**Prompt:"Which classical composer's work does this sound like?"**

| Metric | LTU-AS | | MU-LLaMA | | LLaMA-Adapter | |
|---|---|---|---|---|---|---|
| | Correct | Random | Correct | Random | Correct | Random |
| Precision | 0.333 | 0.337 | 0.226 | 0.235 | 0.281 | 0.241 |
| Recall | 0.285 | 0.288 | 0.618 | 0.642 | 0.506 | 0.433 |
| F1-Score | 0.307 | 0.310 | 0.331 | 0.344 | 0.361 | 0.310 |

**Prompt: "Among {all composers}, whose style does this piece of music sound like?"**

| Metric | LTU-AS | | MU-LLaMA | | LLaMA-Adapter | |
|---|---|---|---|---|---|---|
| | Correct | Random | Correct | Random | Correct | Random |
| Precision | 0.152 | 0.133 | 0.153 | 0.140 | 0.152 | 0.137 |
| Recall | 0.558 | 0.488 | 0.294 | 0.270 | 0.561 | 0.506 |
| F1-Score | 0.238 | 0.209 | 0.201 | 0.185 | 0.239 | 0.216 |

## C.3 SAME NODE EXPERIMENT OF GENRES ON FMA

In addition to the Same Tree experiment on the FMA dataset, we also conducted a Same Node experiment. In this setting, a prediction is considered correct only if the model output label exactly matches the ground-truth label. To further analyze robustness, we evaluated the task under three different prompting strategies (see Section B.2 for details). The results of the Same Node experiment are reported in Table 14.

Table 13: Binary Choices Experiments of Composers on MusicNet

| Prompt: "Between {two composers}, whose style does this piece resemble the most?" | | | |
|---|---|---|---|
| Metric | LTU-AS | MU-LLaMA | LLaMA-Adapter |
| Precision | 0.496 | 0.570 | 0.463 |
| Recall | 0.709 | 0.655 | 0.803 |
| F1-Score | 0.584 | 0.609 | 0.588 |

## D  THE USE OF LARGE LANGUAGE MODELS (LLMS)

We used a large language model extensively as part of our experiments—specifically within the evaluation pipeline that converts free-form model outputs into a canonical representation and related analysis steps—but not for research ideation or manuscript writing. Concretely, we called GPT-4.1-mini (OpenAI) via API, using a rule-based prompt that restricts extraction to explicitly stated labels from a fixed vocabulary. Deterministic post-processing maps variants to canonical forms . For quality control, we audited some random samples per task and resolved disagreements with two human annotators using a simple tie-break rule. We log all API responses to ensure reproducibility and did not transmit private or sensitive data beyond the released benchmarks. LLMs are not authors; the human authors take full responsibility for all content and analyses.

Table 14: Different Prompts of Genre Classification (Same Node) Experiments on FMA

**Prompt: "What is the genre of this song"**

|  | MU-LLaMA | | LLaMA-Adapter | | SALMONN | |
|---|---|---|---|---|---|---|
|  | Correct | Random | Correct | Random | Correct | Random |
| Precision | 0.093 | 0.035 | 0.181 | 0.041 | 0.173 | 0.039 |
| Recall | 0.117 | 0.043 | 0.206 | 0.047 | 0.086 | 0.019 |
| F1-Score | 0.104 | 0.039 | 0.193 | 0.044 | 0.115 | 0.026 |

**Prompt: "What can you infer about the genre of the music"**

|  | MU-LLaMA | | LLaMA-Adapter | | SALMONN | |
|---|---|---|---|---|---|---|
|  | Correct | Random | Correct | Random | Correct | Random |
| Precision | 0.034 | 0.034 | 0.120 | 0.026 | 0.101 | 0.022 |
| Recall | 0.157 | 0.064 | 0.249 | 0.055 | 0.244 | 0.054 |
| F1-Score | 0.108 | 0.044 | 0.162 | 0.036 | 0.143 | 0.031 |

**Prompt: "What genre does this piece of music fall under?"**

|  | MU-LLaMA | | LLaMA-Adapter | | SALMONN | |
|---|---|---|---|---|---|---|
|  | Correct | Random | Correct | Random | Correct | Random |
| Precision | 0.084 | 0.033 | 0.184 | 0.038 | 0.122 | 0.027 |
| Recall | 0.128 | 0.050 | 0.217 | 0.045 | 0.231 | 0.052 |
| F1-Score | 0.102 | 0.039 | 0.199 | 0.041 | 0.160 | 0.036 |