# OpenReview forum: "Factual and Musical Evaluation Metrics for Music Language Models"
_ICLR.cc/2026/Conference — Submitted to ICLR 2026_

### Official Review · Reviewer_ouqf · 2025-10-21

**Soundness:** 2
**Presentation:** 4
**Contribution:** 3
**Rating:** 4
**Confidence:** 4

**Summary:**

The paper demonstrates the limitations of current metrics for evaluating music language models; it then proposes a new metric based on the CLAP audio-text embeddings which more accurately captures the performance of music LMs. Finally the paper proposes a methodology for converting the output of music LMs into a set of labels which can then be evaluated using standard classification or retrieval metrics.

**Strengths:**

The paper addresses a timely and important problem in the music AI literature, which is on the limitations of current (NLP-based) evaluation metrics to evaluate the performance of music language models. The proposed metric based on CLAP embeddings is a simple and yet very effective solution to address this issue. The paper also evaluates a representative range of music LMs using benchmark open datasets.

**Weaknesses:**

* The third and final contribution of this work is relatively disjointed from the first two contributions. At the same time, the paper does not acknowledge the limitations between the "factuality protocol", which appears to only be effective for evaluating performance on a fairly narrow set of multi-label classification tasks related to music (e.g. audio tagging, instrument identification).
* Evaluations in 3.4 only use one variant of CLAP, and could have been expanded as to cover other CLAP variants but also other audio-text embeddings - as to answer the question on whether CLAP is better suited as a model to evaluate music LM performance.
* Section 5 does not include any critical reflections on this work, and does not acknowledge any limitations or areas of possible or future improvement.
* There is no mention in the paper on the possible availability of the evaluation metrics, e.g. as a package or library so that they can be adopted by the community.

**Questions:**

* Section 3.4: I would suggest to include other CLAP variants but also include other common audio-text embeddings in comparative evaluations. This is in order to answer the question on whether CLAPtext is really a good predictor of musicLM performance over other such audio-text models.
* The "factual QA" work in section 4 (I have put the term factual in quotes since I do not believe the term is appropriately used in the paper, given that there are many other possible factual questions not covered by the methodology) converts music LM outputs into a set of tags or labels. This is appropriate for evaluating performance on specific downstream tasks such as audio tagging or instrument identification, however does not capture other important tasks in the wider MIR literature - or new tasks which involve temporal grounding and involve inferences on musical sequences.
* Section 5 can be radically rewritten as to include a more critical and informative discussion on the strengths and areas of development of the proposed methodologies - leading also to possible directions for future work.

---

> ### Author Response · Authors · 2025-11-20
>
> We thank you for your feedback. We appreciate your recognition of the importance of reexamining evaluation practices in music AI and your positive assessment of our use of CLAP-based embeddings as a simple yet effective alternative to traditional NLP metrics. We are also grateful that you found our evaluation across representative music language models and open datasets to be useful and well-motivated. Your comments reinforce the relevance of our contributions to improving evaluation standards in the field.
>
> > Section 3.4: I would suggest to include other CLAP variants but also include other common audio-text embeddings in comparative evaluations. This is in order to answer the question on whether CLAPtext is really a good predictor of musicLM performance over other such audio-text models.
>
> Because our work focuses specifically on the music domain, we opted to use the music-only CLAP checkpoint, as it is better aligned with our research scope than general audio models trained on broader sound datasets. Our goal in Section 3.4 was not to benchmark embedding models exhaustively, but to demonstrate that a well-established music-focused embedding (CLAP with music-only checkpoint) can reveal limitations in current NLP-based metrics. That said, we agree that evaluating alternative audio–text embeddings could further strengthen the analysis, and we would be happy to include such comparisons if the reviewer has recommendations for high-quality embedding models suitable for music understanding.
>
> > The "factual QA" work in section 4 (I have put the term factual in quotes since I do not believe the term is appropriately used in the paper, given that there are many other possible factual questions not covered by the methodology) converts music LM outputs into a set of tags or labels. This is appropriate for evaluating performance on specific downstream tasks such as audio tagging or instrument identification, however does not capture other important tasks in the wider MIR literature - or new tasks which involve temporal grounding and involve inferences on musical sequences.
>
> Our intention is not to restrict “factual” evaluation to only genre, instrumentation, or composer (Appendix C.2), but to introduce a general protocol for assessing factual grounding that can be applied to any music-related attribute, as long as it is possible to define a clear factual target and to prompt the model appropriately. Appendix B.1 outlines prompting rules designed to make the protocol adaptable to a wide range of factual question types. Our current experiments focus on genre, instrumentation, and composer because these are the domains for which high-quality, human-annotated datasets are available. We agree that extending the protocol to tasks involving temporal grounding or richer musical inferences would be valuable, and we would be eager to evaluate these cases if suitable labeled datasets are available.
>
> >The third and final contribution of this work is relatively disjointed from the first two contributions. At the same time, the paper does not acknowledge the limitations between the "factuality protocol", which appears to only be effective for evaluating performance on a fairly narrow set of multi-label classification tasks related to music (e.g. audio tagging, instrument identification).
>
> The third section is intended to diagnose why existing NLP-based metrics fail by showing that they are dominated by lexical similarity rather than factual correctness, which directly motivates the factuality protocol in Section 4. Thus, the two parts are conceptually linked rather than disjoint: the analysis in Section 3 reveals a structural limitation, and the factuality protocol in Section 4 provides a principled way to address it. Regarding the scope of the factuality protocol, we agree that our experiments primarily focus on genre, instrumentation, and composer (appendix C.1) classification. This choice reflects the availability of high-quality, human-annotated datasets rather than a limitation of the protocol itself. As detailed in Appendix B.1, our prompting rules allow the same protocol to be applied to any task where factual attributes can be defined. We would be glad to extend our evaluation to additional musical facets if there are more suitable datasets.

---

> ### Author Response · Authors · 2025-11-20
>
> > Section 5 does not include any critical reflections on this work, and does not acknowledge any limitations or areas of possible or future improvement.
>
> Our work highlights the need for better evaluation methods for open-ended captioning, not only in music but in multimodal generation more broadly. While our factuality protocol provides a strong foundation for structured QA-style evaluation, we acknowledge that it does not directly address fully open-ended captioning. Similarly, CLAPText performed well under the conditions we tested but should not yet be viewed as a definitive solution.
>
> >There is no mention in the paper on the possible availability of the evaluation metrics, e.g. as a package or library so that they can be adopted by the community.
>
> We are currently organizing and expanding our experimental pipeline, and we plan to release a public repository containing the evaluation metrics and code once it is cleaned and documented.

---

### Official Review · Reviewer_S1xe · 2025-10-26

**Soundness:** 2
**Presentation:** 2
**Contribution:** 2
**Rating:** 2
**Confidence:** 3

**Summary:**

This paper focuses on metrics for Music LMs and Factual QA evaluation. It shows that six commonly reported metrics for evaluating Music LMs fail to measure these models’ ability to extract information from audio inputs. The authors also propose a new, musically-aware similarity metric, CLAPText. The experimental results using paraphrase and adversarial text show CLAPText is robust on those tricky cases.

**Strengths:**

- This paper provides a valuable analysis of the music domain, addressing the limitations of commonly used metrics such as BLEU in effectively evaluating models in this context.
- The proposal to use CLAPText as an evaluation metric is a reasonable and intuitive approach. Furthermore, the experimental results with adversarial text offer a suggestive and insightful contribution to the field.

**Weaknesses:**

-	The highest-scoring case (paraphrase) and the lowest-scoring cases (adversarial/random) have been evaluated. However, as a validation of the evaluation metric, it is also necessary to confirm whether diverse cases can be appropriately ranked in order. For example, when there are partial differences (e.g., only some instruments are incorrect, or the information is partially correct but includes additional new incorrect information), the score should change gradually to reflect these differences.
-	While the weaknesses in adversarial cases have been demonstrated, they can be considered special cases. Ideally, the evaluation should enable the comparison between models to determine which one is better. The "Correct" case in Table 2 seems to correspond to the performance evaluation results, but the rankings appear to differ across the metrics. It would be better to verify whether the scores indicated by CLAPText provide more appropriate evaluation values (e.g. the correlation with human evaluation results).

**Questions:**

- The Model columns in the last row of Table 2 are empty, which may look like typos.
- The term "Prompt" in Table 2 is unclear. Wouldn't it be more appropriate to clearly indicate that it replaces the reference used during evaluation, such as "Reference Answer."? (In that case, the "Reference answer" in Figure 1 should be changed to "Gold answer" or something.)
- It would be better to include the actual prompts used for generating paraphrases and adversarial sentences in the Appendix.
- In Table 4, comparing the F1-scores, LlaMA-Adapter seems to perform better than MU-LlaMA. When comparing the evaluation values for the models in MusicQA-Jamendo in Table 2 for the "Correct" case, only ROUGE follows that order. Similarly, for Llama-Adapter and SALMON, Llama-Adapter performs better, but only ROUGE follows that order in any of the data in Table 2. From these results, if Llama-Adapter is a strong model for instrument recognition and genre classification, is it possible that ROUGE evaluates the actual model outputs better than CLAPText?
- It would be better to have some analysis (e.g., human verification) to ensure that the adversarial texts actually contain the expected content.

---

> ### Author Response · Authors · 2025-11-20
>
> We are grateful for your feedback. We appreciate your recognition of our analysis of evaluation challenges in the music domain and your positive assessment of CLAPText as a simple and intuitive metric. We are also grateful that you found the adversarial text experiments to be insightful, as highlighting these weaknesses in existing evaluations is a central goal of our work. Your comments reinforce the value of developing more robust and domain-appropriate evaluation methods for music–language models.
>
> >The highest-scoring case (paraphrase) and the lowest-scoring cases (adversarial/random) have been evaluated. However, as a validation of the evaluation metric, it is also necessary to confirm whether diverse cases can be appropriately ranked in order. For example, when there are partial differences (e.g., only some instruments are incorrect, or the information is partially correct but includes additional new incorrect information), the score should change gradually to reflect these differences.
>
> Our current analysis focuses on the extremes (paraphrased versus adversarial or random) to reveal that most commonly used metrics, with the exception of CLAPText, fail to show statistically significant separation even under these clear conditions. At the same time, our factuality evaluation framework naturally captures intermediate correctness, since partially correct answers directly translate into proportional changes in precision, recall, and F1. Nonetheless, for partially correct answers, we also expect that both n-gram–based metrics (BLEU, ROUGE, etc.) and embedding-based metrics (BERTScore, CIDEr, CLAPText) would exhibit gradual score changes, as they are inherently sensitive to partial overlaps.
>
> > While the weaknesses in adversarial cases have been demonstrated, they can be considered special cases. Ideally, the evaluation should enable the comparison between models to determine which one is better. The "Correct" case in Table 2 seems to correspond to the performance evaluation results, but the rankings appear to differ across the metrics. It would be better to verify whether the scores indicated by CLAPText provide more appropriate evaluation values (e.g. the correlation with human evaluation results).
>
> One of our central findings is that many widely used metrics do not show reliable or consistent differences even between clearly correct and clearly incorrect cases. Because these metrics fail under such controlled conditions, we argue that their model rankings should not be trusted, which is consistent with our observation that rankings vary unpredictably across metrics in Table 2. By contrast, our factuality protocol is explicitly designed to provide a more reliable basis for model comparison by focusing on objective, domain-grounded correctness. We will make this argument more explicit in the revised manuscript.
>
> > The Model columns in the last row of Table 2 are empty, which may look like typos.
>
> The last row of Table 2 corresponds to paraphrase and adversarial variants, which are not outputs from any specific model, hence the Model column is intentionally blank.
>
> > The term "Prompt" in Table 2 is unclear. Wouldn't it be more appropriate to clearly indicate that it replaces the reference used during evaluation, such as "Reference Answer."? (In that case, the "Reference answer" in Figure 1 should be changed to "Gold answer" or something.)
>
> We agree that the terminology in the table may be unclear, and we will revise it to be more precise, along with updating Figure 1 accordingly.
>
> > It would be better to include the actual prompts used for generating paraphrases and adversarial sentences in the Appendix.
>
> The prompts used to generate paraphrases and adversarial outputs (“paraphrase:” and “make minor edits to completely change the meaning”) are shown in Figure 1, and we will additionally include more generated examples in the appendix to provide clearer context.

---

> ### Author Response · Authors · 2025-11-20
>
> > In Table 4, comparing the F1-scores, LlaMA-Adapter seems to perform better than MU-LlaMA. When comparing the evaluation values for the models in MusicQA-Jamendo in Table 2 for the "Correct" case, only ROUGE follows that order. Similarly, for Llama-Adapter and SALMON, Llama-Adapter performs better, but only ROUGE follows that order in any of the data in Table 2. From these results, if Llama-Adapter is a strong model for instrument recognition and genre classification, is it possible that ROUGE evaluates the actual model outputs better than CLAPText?
>
> We do not believe this implies that ROUGE is a superior evaluation metric. ROUGE remains an n-gram–based measure and behaves similarly to BLEU and METEOR. As shown in our paraphrase and adversarial experiments, ROUGE is still strongly influenced by surface-level lexical similarity and fails to reliably distinguish factually incorrect answers from correct ones. Therefore, we do not consider its apparent ranking agreement in isolated cases to be meaningful. In contrast, CLAPText is designed to avoid these lexical traps, and our results show that ROUGE’s rankings should not be interpreted as evidence of better evaluation fidelity.
>
> > It would be better to have some analysis (e.g., human verification) to ensure that the adversarial texts actually contain the expected content.
>
> We will include additional examples of both paraphrased and adversarial outputs in the appendix to increase clarity and transparency and to illustrate that the generated content matches the intended manipulations.

---

### Official Review · Reviewer_cizx · 2025-10-29

**Soundness:** 2
**Presentation:** 2
**Contribution:** 2
**Rating:** 4
**Confidence:** 3

**Summary:**

This paper presents a critical look at evaluation protocols for Music LMs, providing both a better quantitative metric for open-ended QA and an automated evaluation pipeline for factual correctness of open-ended responses.

**Strengths:**

- The authors' motivation is strong, as the current state of evaluation for Music LMs being rather dubious.
- The construction of the factuality evaluation protocol in particular is reasonably strong. Such a framework gets at the heart of the failure modes of many modern Music LMs.

**Weaknesses:**

- It feels as if this paper is caught between two similar yet practically orthogonal contributions: ClapText and their factual evaluation protocol. This muddies the overall flow and contribution of the paper, as substantially more content is dedicated to the weaker results (ClapText) over the correctness evaluation.
- Overall, the evaluation with ClapText is not massively convincing. First off, it is unclear why the authors opted for random sampling of audio prompts rather than Gaussian noise (as done in previous work [1,2]) for their "random" evaluation. The fact that gaussian noise causes prototypical performance degradation is *the point* of using it, as it ensures absolutely no information contained in the audio, while the choice of random audio gives no gaurantee that there isn't information present in the random sample that is relevant to the given query (one could imagine that in a set of reasonably similar songs, this might be more common than you'd think). More importantly, the results show that the ClapText scores are higher for the *adversarial* prompt than for **any** of the "Correct" category. This seems to imply that either (1) ClapText is still sensitive to correctness but the model tested outputs are so bad that they are more wrong than the actively incorrect adversarial prompt or (2) while more robust than the other metrics, ClapText is still fooled by the adversarial responses relative to the possibly *good* outputs of the model when given correct prompts. In the present manuscript, it is thus impossible to tell which case the results are in (as there is no independent evaluation of the tested models' responses), and hence it is hard to claim any strict improvement from the ClapText metric.
- While the pipeline for the factuality evaluation seems strong, the overall evaluation of it is relatively limited. It is unclear to me why the only tested models are ones that have been long since passed by more SOTA systems [2,3]. This makes the evaluation hard to conclude anything from, as it is hard to tell whether the performances shown would continue for more modern systems. This also connects back to the ClapText evaluation, in that using more modern systems (Qwen-2-Audio, Qwen-3-Omni, Audio Flamingo 2, Audio Flamingo 3) would be useful for improving evaluation. While these systems are trained on more general audio rather than just music, they have clear documented improved performance over music-only models on QA tasks [2,3].
- One small point, it would be useful throughout the manuscript to change the reference of "Music LMs" to "Large Audio Language Models (LALMs), as this is the standard nomenclature increasingly being taken in the community [1,2], and "MusicLMs" runs the risk of confusing readers with the series of music *generation* models that using AR transformers (including the one expliciltly called MusicLM [4, 5]).

Overall, I think there are too many issues in the present manuscript to recommend acceptance, but I do believe the present draft could be reasonably improved by expanding the evaluation suite to use more modern models and performing a more careful analysis of the ClapText metric.

[1] Kumar, Sonal, et al. "Mmau-pro: A challenging and comprehensive benchmark for holistic evaluation of audio general intelligence." arXiv preprint arXiv:2508.13992 (2025).
[2] Zang, Yongyi, et al. "Are you really listening? boosting perceptual awareness in music-qa benchmarks." arXiv preprint arXiv:2504.00369 (2025).
[3] Weck, Benno, et al. "Muchomusic: Evaluating music understanding in multimodal audio-language models." arXiv preprint arXiv:2408.01337 (2024).
[4] Agostinelli, Andrea, et al. "Musiclm: Generating music from text." arXiv preprint arXiv:2301.11325 (2023).
[5] Copet, Jade, et al. "Simple and controllable music generation." Advances in Neural Information Processing Systems 36 (2023): 47704-47720.

**Questions:**

See weaknesses.

---

> ### Author Response · Authors · 2025-11-20
>
> We thank you for your thoughtful feedback. We appreciate your recognition of the motivation behind our work, especially given the current challenges and inconsistencies in evaluating Music LMs. We are grateful that you found the design of our factuality evaluation protocol to be a strong component of the paper, and we are encouraged by your observation that our framework directly targets core failure modes of modern music-language models.
>
> > It feels as if this paper is caught between two similar yet practically orthogonal contributions: ClapText and their factual evaluation protocol. This muddies the overall flow and contribution of the paper, as substantially more content is dedicated to the weaker results (ClapText) over the correctness evaluation.
>
> We appreciate your perspective and apologize for any confusion caused by the current framing. Our intention is not to present ClapText and the factuality protocol as independent or competing contributions, but rather as sequential components of a single narrative. ClapText was explored as an alternative metric precisely to probe whether existing approaches are overly influenced by lexical similarity. The key contribution of Part 3 is therefore not the metric itself, but the comparative experimental protocol demonstrating that many current metrics fail to distinguish factual correctness from surface-level similarity, a limitation that motivates the factuality protocol introduced in Part 4. These two components are thus closely connected rather than orthogonal. We recognize that this relationship may not have been sufficiently clear in the introduction, and we will revise the framing to better highlight the intended flow of the contributions.
>
> > Overall, the evaluation with ClapText is not massively convincing. First off, it is unclear why the authors opted for random sampling of audio prompts rather than Gaussian noise (as done in previous work [1,2]) for their "random" evaluation. The fact that gaussian noise causes prototypical performance degradation is *the point* of using it, as it ensures absolutely no information contained in the audio, while the choice of random audio gives no gaurantee that there isn't information present in the random sample that is relevant to the given query (one could imagine that in a set of reasonably similar songs, this might be more common than you'd think).
>
> Our rationale is that current multimodal music–language models are not built to interpret purely noisy or corrupted signals, so evaluating them on white noise may not meaningfully reflect their factual reasoning capabilities. A central finding of our work is that several metrics reward responses that sound like music analysis regardless of factual correctness, and therefore contrasting correct audio with random but still musical signals provides a more realistic and informative baseline for exposing this failure mode. While we acknowledge the tradition of noise-based baselines, we believe random musical inputs better reflect the kinds of off-distribution yet structured audio models actually encounter. We are open to including noise-based controls if the reviewers feel these experiments would materially strengthen the paper, and we will clarify this design decision more explicitly in the revision.
>
> > More importantly, the results show that the ClapText scores are higher for the *adversarial* prompt than for **any** of the "Correct" category. This seems to imply that either (1) ClapText is still sensitive to correctness but the model tested outputs are so bad that they are more wrong than the actively incorrect adversarial prompt or (2) while more robust than the other metrics, ClapText is still fooled by the adversarial responses relative to the possibly *good* outputs of the model when given correct prompts. In the present manuscript, it is thus impossible to tell which case the results are in (as there is no independent evaluation of the tested models' responses), and hence it is hard to claim any strict improvement from the ClapText metric.
>
> We agree that CLAPText can be fooled by adversarial responses, but all models and metrics are susceptible to carefully constructed adversarial inputs. However, our experiments show that CLAPText still provides useful discriminatory power on non-adversarial model outputs, where traditional NLP metrics largely collapse to measuring only surface-level lexical similarity. The goal of our evaluation is not to claim that CLAPText is immune to adversarial manipulation, but rather that it behaves more robustly than existing metrics under standard conditions and reflects meaningful differences across model outputs that BLEU, ROUGE, and similar metrics fail to capture. We will clarify this point in the revised manuscript.

---

> ### Author Response · Authors · 2025-11-20
>
> > While the pipeline for the factuality evaluation seems strong, the overall evaluation of it is relatively limited. It is unclear to me why the only tested models are ones that have been long since passed by more SOTA systems [2,3]. This makes the evaluation hard to conclude anything from, as it is hard to tell whether the performances shown would continue for more modern systems. This also connects back to the ClapText evaluation, in that using more modern systems (Qwen-2-Audio, Qwen-3-Omni, Audio Flamingo 2, Audio Flamingo 3) would be useful for improving evaluation. While these systems are trained on more general audio rather than just music, they have clear documented improved performance over music-only models on QA tasks [2,3].
>
> We have conducted additional experiments with Audio Flamingo 3. The new findings continue to support our central conclusion: existing NLP-based metrics remain strongly influenced by lexical similarity and often fail to capture true factual correctness, even for newer high-performing models. At the same time, the factuality protocol reveals encouraging improvements that modern models indeed demonstrate higher factual consistency than older music-only systems.
>
> |              | Audio Flamingo 3 (MusicQA-Jamendo) |               |
> | ------------ | ---------------------------------- | ------------- |
> |              | Correct Song                       | Wrong Song    |
> | BLEU Score   | 0.1993334783                       | 0.1826284838  |
> | BLEU-4 Score | 0.1010164235                       | 0.09269190722 |
> | METEOR Score | 0.1580055549                       | 0.1545424156  |
> | ROUGE Score  | 0.168165289                        | 0.1596736039  |
> | BERTScore    | 0.883143127                        | 0.8813924193  |
> | CIDEr        | 0.09151304966                      | 0.07782552839 |
> | CLAPText     | 0.472486                           | 0.444394      |
>
> |           | Audio-Flamingo 3 (Factuality-MusicNet)                       |            |
> | --------- | ------------------------------------------------------------ | ---------- |
> |           | Prompt: "Which instruments are used in this piece of music?" |            |
> |           | Correct Song                                                 | Wrong Song |
> | Precision | 0.692                                                        | 0.376      |
> | Recall    | 0.767                                                        | 0.417      |
> | F1-Score  | 0.728                                                        | 0.396      |
> |           | Prompt: "Among {all instruments}, which instruments are used in this piece of music?" |            |
> |           | Correct Song                                                 | Wrong Song |
> | Precision | 0.236                                                        | 0.193      |
> | Recall    | 0.881                                                        | 0.721      |
> | F1-Score  | 0.372                                                        | 0.305      |
>
> From the new experiment, we observe that AF3, as one of the current sota models, achieves the strongest performance by a large margin on our factuality protocol, while the results from traditional NLP-based metrics do not reflect this improvement. A likely explanation is that AF3 is able to comment on a wider range of musical aspects, such as patterns or tempo, whereas older models, although also different from the ground truth, often produce responses that are musically irrelevant or nonsensical. Because traditional NLP metrics primarily capture lexical similarity, they treat these two types of errors as equally distant from the ground truth, leading to the misleading impression that AF3 is “no better” than older models. Our factuality protocol, by contrast, is able to surface these meaningful qualitative differences in model behavior. Here is an example of AF3’s output and LTU-AS’s output to a free-form question:
>
> Q: Describe the audio
>
> GT: The audio is a dance track with elements of electronic, house, and techno music.
>
> AF3: Electronic music with a fast tempo and repetitive synth patterns, creating an energetic and intense atmosphere.
>
> LTU-AS: The speaker is talking about having fun and enjoying music while someone else plays it loudly in the background. The background sounds of electronic music suggest that this may be happening at a party or club where people are dancing and having a good time. The speaker seems to be enthusiastic and excited about the music.

---

> ### Author Response · Authors · 2025-11-20
>
> > One small point, it would be useful throughout the manuscript to change the reference of "Music LMs" to "Large Audio Language Models (LALMs), as this is the standard nomenclature increasingly being taken in the community [1,2], and "MusicLMs" runs the risk of confusing readers with the series of music *generation* models that using AR transformers (including the one expliciltly called MusicLM [4, 5]).
>
> We agree that adopting the increasingly standard term “Large Audio Language Models (LALMs)” helps avoid confusion with autoregressive music-generation systems, including the MusicLM family. We will update the terminology throughout the revised manuscript to align with community conventions.

---

### Official Review · Reviewer_pLNp · 2025-10-31

**Soundness:** 1
**Presentation:** 2
**Contribution:** 2
**Rating:** 2
**Confidence:** 4

**Summary:**

The paper argues that common text metrics (BLEU, METEOR, BERTScore) fail to assess whether Music LMs actually use audio. It proposes (i) CLAPText, a music-aware text similarity metric, and (ii) a factual QA protocol that converts open-ended responses into structured labels to compute precision/recall/F1. Experiments on MusicQA-style data and FMA/MusicNet tasks suggest text metrics cannot distinguish correct-audio from random-audio baselines, while the proposed metrics better reflect audio grounding.

**Strengths:**

Clear problem framing. Highlights a real gap: NLP metrics reward fluency, not audio-grounded correctness.

Simple, reusable tools. CLAPText is a practical drop-in; the factual QA pipeline is modality-agnostic.

Baseline design is insightful. Random-audio and adversarial paraphrase conditions stress-test audio use vs. surface text overlap.

Open datasets/code intent. Encourages reproducibility and broader adoption.

**Weaknesses:**

Questionable data quality for MusicQA. A large portion of prompts/answers stems from MPT-7B generation; prior work reports hallucinations and low musician approval (e.g., MusiLingo) — weakening conclusions about “correct” vs. “random” gaps when “correct” itself is noisy.

Model coverage is thin. Aside from SALMONN, evaluated models are not the strongest current baselines; conclusions about metric validity would be more convincing with Qwen-Audio, Qwen2.5-Omni, ChatGPT-5, Gemini 2.5 Pro, etc.

Adversarial answers lack expert validation. No musician audit to confirm that “adversarial” outputs are truly incorrect yet linguistically plausible.

Narrow attribute set. Factual QA focuses on genre and instrument; key musical facets (emotion/mood, tempo, key/tonality, meter, timbre attributes) are omitted, limiting external validity.

Potential prompt sensitivity. Results may hinge on prompting; robustness across prompts and extraction rules is under-analyzed.

**Questions:**

Quality control of “correct” references.
What fraction of MusicQA items were musician-verified? Please report inter-annotator agreement and an error taxonomy (hallucination, ambiguity, label noise). How do results change when restricting to a high-quality subset?

Stronger baselines.
Can you include evaluations of Qwen-Audio / Qwen2.5-Omni / ChatGPT-5 / Gemini 2.5 Pro to test whether CLAPText and the factual protocol still separate correct-audio from random-audio with state-of-the-art models?

Adversarial validation.
Will you run a blind musician audit on adversarial answers to confirm factual incorrectness and estimate the false-negative rate of your protocol?

Attribute coverage.
Can you extend factual QA to tempo (BPM ranges), key/scale, meter, dynamic markings, and emotion/mood labels? Even small, well-vetted subsets would strengthen claims of generality.

Noise/no-audio controls.
Please add no-audio, shuffled-audio, and additive-noise baselines to quantify how CLAPText and factual F1 degrade with audio corruption.

Prompt and extractor robustness.
How sensitive are results to (a) prompt templates and (b) the keyword-extraction LLM? Consider reporting variance across prompts and an ablation using rule-based extractors.

---

> ### Author Response · Authors · 2025-11-20
>
> Thank you for your feedback. We sincerely appreciate your recognition of our contribution in identifying the gap between existing NLP metrics and the factuality that users expect from music language models, as well as your praise for the insightfulness of our baseline design. We are also grateful for your positive comments on CLAPText, noting its simplicity and reusability.
>
>
>
> > Quality control of “correct” references. What fraction of MusicQA items were musician-verified? Please report inter-annotator agreement and an error taxonomy (hallucination, ambiguity, label noise). How do results change when restricting to a high-quality subset?
>
> We acknowledge your concern regarding the quality of MusicQA. Unfortunately, there is currently a limited supply of publicly available music–text datasets with free-form, human-written descriptions; most existing paired datasets in this domain, including those used in prior work, rely on LLM-generated captions or expansions from metadata due to the scarcity of annotated musical text. MusicQA remains one of the widely used and established benchmarks in the community, and we adopt it for consistency and comparability with prior work. We agree that higher-quality human-written datasets would be valuable, and we would be grateful if the reviewer could suggest alternative datasets that better align with the goals of this evaluation.

---

> ### Author Response · Authors · 2025-11-20
>
> > Stronger baselines. Can you include evaluations of Qwen-Audio / Qwen2.5-Omni / ChatGPT-5 / Gemini 2.5 Pro to test whether CLAPText and the factual protocol still separate correct-audio from random-audio with state-of-the-art models?
>
> We would like to clarify that the primary focus of our work is on evaluating metrics, not benchmarking models, so our conclusions do not rely on exhaustively covering sota systems. That said, we agree that including stronger contemporary models helps contextualize the evaluation landscape. In line with this, we have conducted additional experiments on Audio Flamingo 3, and these new results will be incorporated into the revised version of the paper. Importantly, they continue to support our central finding that currently used NLP-based metrics remain dominated by lexical similarity, even for newer high-performing models.
>
> |              | Audio Flamingo 3 (MusicQA-Jamendo) |               |
> | ------------ | ---------------------------------- | ------------- |
> |              | Correct Song                       | Wrong Song    |
> | BLEU Score   | 0.1993334783                       | 0.1826284838  |
> | BLEU-4 Score | 0.1010164235                       | 0.09269190722 |
> | METEOR Score | 0.1580055549                       | 0.1545424156  |
> | ROUGE Score  | 0.168165289                        | 0.1596736039  |
> | BERTScore    | 0.883143127                        | 0.8813924193  |
> | CIDEr        | 0.09151304966                      | 0.07782552839 |
> | CLAPText     | 0.472486                           | 0.444394      |
>
> |           | Audio-Flamingo 3 (Factuality-MusicNet)                       |            |
> | --------- | ------------------------------------------------------------ | ---------- |
> |           | Prompt: "Which instruments are used in this piece of music?" |            |
> |           | Correct Song                                                 | Wrong Song |
> | Precision | 0.692                                                        | 0.376      |
> | Recall    | 0.767                                                        | 0.417      |
> | F1-Score  | 0.728                                                        | 0.396      |
> |           | Prompt: "Among {all instruments}, which instruments are used in this piece of music?" |            |
> |           | Correct Song                                                 | Wrong Song |
> | Precision | 0.236                                                        | 0.193      |
> | Recall    | 0.881                                                        | 0.721      |
> | F1-Score  | 0.372                                                        | 0.305      |
>
> From the new experiment, we observe that AF3, as one of the current sota models, achieves the strongest performance by a large margin on our factuality protocol, while the results from traditional NLP-based metrics do not reflect this improvement. A likely explanation is that AF3 is able to comment on a wider range of musical aspects, such as patterns or tempo, whereas older models, although also different from the ground truth, often produce responses that are musically irrelevant or nonsensical. Because traditional NLP metrics primarily capture lexical similarity, they treat these two types of errors as equally distant from the ground truth, leading to the misleading impression that AF3 is “no better” than older models. Our factuality protocol, by contrast, is able to surface these meaningful qualitative differences in model behavior. Here is an example of AF3’s output and LTU-AS’s output to a free-form question:
>
> Q: Describe the audio
>
> GT: The audio is a dance track with elements of electronic, house, and techno music.
>
> AF3: Electronic music with a fast tempo and repetitive synth patterns, creating an energetic and intense atmosphere.
>
> LTU-AS: The speaker is talking about having fun and enjoying music while someone else plays it loudly in the background. The background sounds of electronic music suggest that this may be happening at a party or club where people are dancing and having a good time. The speaker seems to be enthusiastic and excited about the music.

---

> ### Author Response · Authors · 2025-11-20
>
> > Adversarial validation. Will you run a blind musician audit on adversarial answers to confirm factual incorrectness and estimate the false-negative rate of your protocol?
>
> MusicQA is constructed from human-curated labels originating from datasets such as MusicCaps, MagnaTagATune, and Jamendo, so replacing these human-generated keywords during adversarial construction already pushes the answers away from what listeners would consider correct while preserving linguistic plausibility. To further increase transparency, we will add five representative examples of adversarial paraphrases to the appendix, illustrating how they differ from the original ground truth.
>
> Some examples of Ground Truth compared to its adversarial paraphrase can be shown here:
>
> GT: This audio is a **rock song** with elements of **acoustic guitar, bass, drums, and electric guitar**.
>
> AD: This audio is a classical piece with elements of **piano, violin, cello, and flute**.
>
>
>
> GT: In the audio, you can hear a **soft piano melody, gentle percussion, and acoustic guitar chords**. The synthesizer adds a **calm**, **ambient** feel to the track.
>
> AD: In the audio, you can hear a **funky bassline, driving drums, and electric guitar riffs**. The synthesizer adds a **groovy**, **electronic** feel to the track.
>
>
>
> GT: From the audio, we can infer that the composer has a **deep** understanding of classical music and has **taken great care** to create a piece that is both beautiful and emotionally resonant.
>
> AD: From the audio, we can infer that the composer has a **limited** understanding of classical music and has **neglected** to create a piece that is both beautiful and emotionally resonant.
>
>
>
> > Attribute coverage. Can you extend factual QA to tempo (BPM ranges), key/scale, meter, dynamic markings, and emotion/mood labels? Even small, well-vetted subsets would strengthen claims of generality.
>
> We agree that expanding factual evaluation to additional musical facets would be valuable. Our current protocol focuses on genre and instrumentation primarily due to data limitations: high-quality, human-annotated datasets covering mood, timbre, key, meter, and other attributes are scarce. Another experiment we have conducted on classical composers of MusicNet dataset is shown in appendix C.2 in the supplementary material, but we considered that the relative information may not be well represented in the training data of the models, so these questions might be too hard to be used as an effective benchmark in testing the models ability in understanding music. We also note that many of these attributes, such as mood and timbre, are inherently subjective and less standardized, whereas genre and instrumentation provide more objective and widely agreed-upon labels for evaluating factual consistency. We would be grateful if you could recommend high-quality datasets in these broader domains, and we view this as an important direction for future work.
>
>
>
> > Noise/no-audio controls. Please add no-audio, shuffled-audio, and additive-noise baselines to quantify how CLAPText and factual F1 degrade with audio corruption.
>
> We understand your suggestion, and acknowledge that noise-based controls are commonly used in prior work. We do believe that current multimodal music–language models are not designed to interpret purely noisy or corrupted audio, so evaluating them on non-music signals may not meaningfully reflect their factual reasoning capability. One of our key findings is that several metrics reward answers that appear to analyze music rather than those that are factually correct; therefore, we think comparing correct music against random but still musical signals provides a more informative and realistic baseline for this phenomenon. At the same time, we recognize the motivation behind noise-based baselines and are open to including them if the reviewers feel they would meaningfully strengthen the paper. We will clarify this design choice in the revision, and we appreciate your perspective.
>
>
>
> > Prompt and extractor robustness. How sensitive are results to (a) prompt templates and (b) the keyword-extraction LLM? Consider reporting variance across prompts and an ablation using rule-based extractors.
>
> As discussed in Appendix C.1, we evaluated multiple prompt variants with different formats and linguistic styles to assess robustness. The version reported in the main paper is the one that performed best on average across models. We will clarify this evaluation procedure more explicitly and ensure that the appendix highlights the consistency of results across prompt variants.

---

> ### Comment · Reviewer_pLNp · 2025-11-20
>
> Thank you for the additional clarifications. After reviewing the responses, I’m keeping the original score. Key concerns, such as the reliance on noisy MusicQA data, limited model coverage, lack of adversarial validation by experts, and narrow attribute coverage, remain unresolved. While the new experiments are valuable, they don’t fully address the core weaknesses of the paper.

---

### Meta-Review · Area_Chair_BQ2M · 2026-01-06

**Summary:**

The reviewers' concerns are
- Ground-truth / dataset noise makes “factual correctness” hard to trust
- Not enough models in test
- Factual protocol is narrow / not truly “factual” in the broader MIR sense; limited attributes and missing expert validation for adversarial

**Reviewer Concerns:**

- Ground-truth / dataset noise makes “factual correctness” hard to trust

Not resolved. Authors acknowledge the limitation but do not provide musician-verified subsets, agreement statistics, or noise-filtered re-analysis; thus core validity concern remains. In my personal opinion, the quality of the data/label is the life of a benchmark. So this concern is valid and need to be solved.

- Not enough models in test

Partially. The authors added Audio Flamingo 3 results but other requested coverage (e.g., Qwen-Audio/Qwen2.5-Omni/Gemini) is still missing.

- Factual protocol is narrow / not truly “factual” in the broader MIR sense; limited attributes and missing expert validation for adversarial

Not resovled

**Reviewer Scores:**

I believe the reviewers will keep their original rating.

---

### Decision · Program_Chairs · 2026-01-26

Reject